# Characterization of *Pediococcus acidilactici* PFC69 and *Lactococcus lactis* PFC77 Bacteriocins and Their Antimicrobial Activities in Tarhana Fermentation

**DOI:** 10.3390/microorganisms8071083

**Published:** 2020-07-21

**Authors:** Halil İbrahim Kaya, Ömer Şimşek

**Affiliations:** 1Department of Food Engineering, University of Bayburt, 69000 Bayburt, Turkey; halilkaya@bayburt.edu.tr; 2Department of Food Engineering, University of Pamukkale, 20017 Denizli, Turkey

**Keywords:** lactic acid bacteria, bacteriocin as biopreservatives, antimicrobial interventions, cereal fermentation

## Abstract

Tarhana is a traditional cereal product fermented by lactic acid bacteria (LAB) and yeast strains that has gained special interest recently as an infant nutrition. Tarhana contains wheat flour, yogurt, and various vegetables that might create a microbiological toxicological risk, especially for *Bacillus cereus* and *Staphylococcus aureus*. In this study, characterization of the metabolites responsible for antibacterial activity of *Pediococcus acidilactici* PFC69 and *Lactococcus lactis* PFC77 strains obtained from tarhana was performed, and antibacterial effects were detected against *B. cereus* ATCC 11778 and *S. aureus* ATCC 29213 during the fermentation. A total of 12,800 AU/mL antibacterial activity was observed for the supernatants of the PFC69 and PFC77 strains that were found to be stable at high temperature and in low pH conditions and sensitive to proteases, suggesting the antimicrobial metabolite is a bacteriocin. These bacteriocins were further purified and their molecular sizes were determined as 4.5 and 3.5 kDa, respectively. Importantly, inoculation of PFC69 and PFC77 to tarhana dough significantly decreased *B. cereus* ATCC 11778 and *S. aureus* ATCC 29213 amounts from the fifth day of fermentation compared to the control dough samples. *P. acidilactici* PFC69 and *L. lactis* PFC77 strains were concluded as bioprotective cultures for tarhana and these strains were offered for other cereal-based fermentations.

## 1. Introduction

Traditional fermented foods are a valuable resource for providing cheap, practical, and useful foods in the modern world. Tarhana is a traditional fermented product obtained by the fermentation process of a dough prepared with wheat flour, yogurt, yeast, and various vegetables and spices (tomato, red pepper, onion, peppermint, salt) that is then subjected to drying and grinding. This product is mainly produced and consumed as instant soup in Anatolia, but similar products have also been produced in Greece, Hungary, and Egypt with different names [1]. Due to its rich composition including plant and animal nutritional sources, tarhana has a nutritious feature, in addition, due to its easy digestive properties, it can be the first food after breast milk in infant nutrition as well as an alternative nutrient-rich food used in the feeding of patients [2]. However, the microbiological safety of tarhana is critical, especially for individuals with weak immunity, but also for healthier consumers.

Tarhana dough has a complex fermentation microbiota where yeast is found together with lactic acid bacteria (LAB). LAB increases the acidity with organic acid production, and yeasts are mainly responsible for the leavening and aromatic development of the dough with the production of CO_2_ and alcohol. Even though the microbial diversity is high at the beginning of fermentation, acid-tolerant yeasts form the dominant flora with LAB in later flora [3,4,5].

In addition to their contribution for improving structural and aromatic properties in fermented food products, LAB may protect the product against microbial degradation or risks of microbial contamination through various antibacterial metabolites they may produce. In previous studies, it has been reported that LAB have antibacterial effects due to the lactic acid, H_2_O_2_, diacetyl, and most importantly bacteriocin production [6]. Bacteriocins are peptides or protein-nature metabolites that are synthesized by the bacteria ribosomally, secreted into the environment, and are generally effective in the inhibition of close relative species [7]. In studies performed up to date, bacteriocins have been found to have significant antibacterial effects against many pathogenic bacteria including *Bacillus cereus*, *Listeria monocytogenes*, and *Staphylococcus aureus* [8,9].

*Bacillus cereus* is a common soil saprophyte and has an increasing importance as an opportunistic pathogen. Consumption of food contaminated with *B. cereus*, considered as a foodborne pathogen, can cause food poisoning. Some food products, in particular, have a greater risk of *B. cereus* contamination such as cereals, starchy food ingredients, meat and milk products as well as dry foods and spices [10]. *S. aureus* is one of the most important factors of foodborne diseases in humans. Some strains are resistant to high concentrations of salt as well as against some antibiotics. It has been determined that humans are the most important factor in the contamination of *S. aureus*, which causes food poisoning. It is commonly found in foods and plant surfaces. Both of these bacteria are the cause of many food poisonings frequently reported and are seriously considered in food safety [11].

It is known that pathogens such as *B. cereus* and *S. aureus* can survive in the early days of fermentation, although the amount of acidity gradually increases during the fermentation of tarhana dough. However, the toxins they produce can negatively affect human health, and even if the vast majority of these pathogens die, their active toxins may still persist in the food systems [12]. Therefore, the rapid cessation of these species in the early days of fermentation will allow safe tarhana production and thus result in the protection of individuals and infants. Within this context, in this study, the characterization of the bacteriocins produced by *P. acidilactici* PFC69 and *L. lactis* PFC77 strains previously isolated from tarhana dough was performed and then the antibacterial effects against *B. cereus* ATCC 11778 and *S. aureus* ATCC 29213 were detected during the fermentation of tarhana dough.

## 2. Materials and Methods

### 2.1. Microorganisms and Growth Conditions

LAB isolates originating from tarhana dough and the indicator strains (Table 1) used for antimicrobial activity were supplied from Pamukkale University, Food Engineering Department Culture Collection (PUFECC, WDCM 1019). Fifty different vancomycin-resistant *Enterococcu faecium* (VRE) and methicillin-resistant *Staphylococcus aureus* (MRSA) isolates were kindly provided by the Behçet Uz Children’s Hospital located in İzmir-Turkey. Indicator strains were grown in liquid media for 18 h at 30 or 37 °C in brain heart infusion (BHI, Merck, Darmstadt, Germany) and Luria-Bertani (LB, Merck, Darmstadt, Germany); LAB strains were gown at 30 °C for 18 h in de-Man, Rogosa, and Sharpe (MRS-Merck, Darmstadt, Germany) and M17 glucose (M17G 0.5% glucose-Merck, Darmstadt, Germany) liquid or solid media including 2% agar (MRS-Merck, Darmstadt, Germany). All strains were stored at −80 °C in glycerol with a final concentration of 30% and were used after being propagated twice in the appropriate medium.

### 2.2. Determination of the Antimicrobial Spectrums of Lactic Acid Bacteria (LAB) Isolates

Agar spot and well diffusion tests were performed to identify bacteriocin production capabilities of *P. acidilactici* PFC69 and *L. lactis* PFC77 for antibacterial activity. In the agar spot method, LAB strains were spread to MRS and M17G agar plates and incubated at 30 °C for 24 h. Indicator bacteria were grown in LB and BHI broths and inoculated to 5 mL of soft agar (LB and BHI) containing 0.7% agar that was poured onto MRS and M17G agar plates as a second layer and spread homogeneously. These agar plates were then left to incubate for 18–24 h at the appropriate temperature for the growth of indicator bacteria. At the end of this period, the inhibition zone diameters formed by the strains against indicator bacteria were recorded [13].

For the determination of the antibacterial activity by the well diffusion method, isolates grown for 18 h at 30 °C were centrifuged at 10,000× *g* for 15 min (Hettich, Universal 30 RF, Kirchlengern, Germany). The supernatants were transferred to new tubes and neutralized by adjusting the pH to 6 with 6 N NaOH and sterilized by passing it through a 0.45 µm pore diameter membrane filter (Merck-Millipore, Burlington, MA, USA). Indicator bacteria grown in LB and BHI liquid medium were homogeneously spread by pouring into a second layer of nutrient agar (NA, Merck, Darmstadt, Germany) plates by inoculating into 0.7 mL of soft LB and BHI agar medium containing 0.7% agar. Five-millimeter diameter wells were formed on the NA plates and then 100 µL supernatants were filled into the wells. After incubation, the formation of inhibition zones around the wells were examined and recorded [14].

### 2.3. Determination of pH, Heat, and Enzyme Sensitivity of Antimicrobial Activity of Supernatants

The pH values of the culture supernatant of bacteriocin producer LAB were adjusted between 2.0–11.0 using 6 N NaOH or 6 N HCl. These supernatants were sterilized by passing it through 0.45 µm pore diameter membrane filters. The sterilized samples were kept at +4 °C for 24 h. For the control, untreated culture supernatants were used. For the determination of the effect of temperature on antimicrobial activity, heat treatment in a water bath (Memmert, WNB 14, Darmstadt, Germany) at 80, 90, and 100 °C for 5, 10, and 15 min, and in an autoclave (Hirayama, HV-50L, Japan) at 121 °C for 15 min for the prepared media was conducted and then samples from these media were tested against indicator bacteria. Culture supernatants without heat treatment were used as the control. The effect of enzymes on neutralized culture supernatants were determined by applying 1 mg/mL trypsin (pH 7.0, Merck, Darmstadt, Germany), α-chymotrypsin (pH 7.0, Sigma, St. Louis, MI‎, USA), proteinase K (pH 7.0, Sigma, USA), pepsin (pH 3.0, Sigma, USA), α-amylase (pH 7.0, Sigma, USA), lipase (pH 7.0, Sigma, USA), catalase (pH 7.0, Sigma, USA), and lysozyme (pH 7.0, Sigma, USA) for 2 h of incubation at 30 and 37 °C. Subsequently, enzyme activities were terminated by 5 min heat treatment at 100 °C. After pH, temperature, and enzyme applications, bacteriocin activities were calculated as the arbitrary unit (AU) obtained by the critical dilution method multiplied by the highest dilution rate received by 1000/transferred amount [15].

### 2.4. Production and Purification of Bacteriocins from LAB Strains

The *P. acidilactici* PFC69 and *L. lactis* PFC77 strains were grown in 500 mL MRS and M17G liquid medium for 18 h incubation at 30 °C. Afterward, the supernatant was separated with 12,500× *g* for 30 min. Proteins were precipitated with 60% ammonium sulfate from the culture supernatant for overnight at +4 °C with gentle shaking (WiseShake, SHO-1D, BK, ‎Kansas City, MI, USA). The precipitated proteins were collected with centrifugation at +4 °C, 15,300× *g* for 60 min. Subsequently, the supernatant was poured and the precipitate was dissolved in 5 mM sodium phosphate (pH 6.0) buffer and passed through strata C18-E (Phenomenex, SPE, 5 g, 20 mL, Torrance, CA, USA) column. Bacteriocins were recovered with 70% isopropanol and 0.1% TFA, which were further purified and collected with reverse phase UHPLC system (UltiMate 3000, Thermo-Fisher Scientific, Waltham, MA, USA) equipped with a C18 (Nucleosil, Supelco, Bellefonte, PE, USA) column, DAD detector, and fraction collector (Thermo-Fisher Scientific, USA) under gradient conditions using 0.1% TFA (Solvent A) and 60% acetonitrile (Solvent B) containing 0.1% TFA at 1 mL flow rate. The gradient program was 0–5 min 70% A and 30% B, 5–40 min 30% A and 70% B, 40–50 min 100% B, and 50–60 min 100% A.

### 2.5. Determination of Molecular Size of Bacteriocins

The molecular size of bacteriocins was determined using tricine-sodium dodecyl sulfate-polyacrylamide gel electrophoresis (Tricine-SDS-PAGE) [16]. The Bio-Rad (Mini-PROTEAN Tetra Cell, Hercules, CA, USA) gel preparation system was used in tricine-SDS-PAGE application. A current of 75 V for migration in the agglomerated gel and one of 100 V for migration in the separating gel were applied and the gel was subjected to staining application. Molecular sizes of the proteins were determined compared to the relative mobility (Rf) values of the marker proteins (Thermo, Spectra Multicolor Low Range, Waltham, MA, USA).

### 2.6. Screening Bacteriocin Genes in Bacteriocin Producers

A PCR screening of the partial sequences of the structural genes in the genomes of two bacteriocin producers was applied in order to identify the bacteriocins produced. The genomic DNA of *P. acidilactici* PFC69 and *L. lactis* PFC77 were purified by using a genomic DNA purification kit (Invitrogen, Thermo-Fisher Scientific, USA). The 22 primer pairs designed previously according to the registered bacteriocin structural gene sequences and reported by Macwana and Muriana [17] were utilized in PCR in a total volume of 40 µL including 4 µL master mix (SOLIS/Bio Dyne, 5* FIREPolR, Tartu, Estonia), 1 µL primer, and 2 µL genomic DNA. The PCR program was initial denaturation at 95 °C for 5 min, 30 cycles of 95 °C for 30 s, 55 °C for 30 s, 72 °C for 1 min, and 72 °C for 10 min final extension. The amplified PCR products were sequenced and searched in the NCBI (National Center for Biotechnology Information Gene Bank) database for similarity analyses.

### 2.7. Tarhana Production, Microbiological, and Chemical Analyses in Tarhana Dough Fermentation

Tarhana ingredients (red pepper, green pepper, and onion) were initially chopped after being washed in a blender (Arzum, Prokit 444, İstanbul, Turkey) that was subsequently mixed with yogurt (Pınar Co. İzmir, Turkey) and wheat flour (Type 550, Söke Co. Aydın, Turkey), resulting in 7 kg of semi-solid tarhana dough. This master dough was then divided into 1 kg of seven portions. One was used as the control (C); two doughs were inoculated with only indicator strains (*B. cereus* ATCC 11778 and *S. aureus* ATCC 29213 separately) coded as B and S; two doughs were inoculated with *P. acidilactici* PFC69 and one indicator strain, coded as BP and SP; and two doughs were inoculated with *L. lactis* PFC77 and one indicator strains, coded as BL and SL. All the bacteriocin producer and indicator strains were inoculated at 10^7^ and 10^4^ cfu/g to the relevant dough samples, respectively. All the prepared tarhana samples were fermented at 21 ± 1 °C for 21 days separately.

The microbiological analyses were performed at 0, 1, 3, 5, 10, 15, and 21 days of fermentation. The LAB amount was enumerated on MRS-5C medium containing 0.01% cycloheximide after 48 h incubation at 30 °C. Total aerobic mesophilic bacteria (TAMB) and yeasts were determined on the Plate Count Agar (PCA, Merck, Germany) and Dichloran Rose Bengal Chloramphenicol agar (DRBC Agar, Merck, Germany) after 48 h of incubation at 30 °C. *S. aureus* was detected on Baird Parker Agar (BPA Agar, Merck, Germany) medium after 24 h incubation at 37 °C, whereas Chromogenic Bacillus Cereus Selective agar (Oxoid, Columbia, MD, USA) supplemented with polymyxin B and trimethoprim (Oxoid, USA) was used to for *Bacillus cereus*, forming blue-green colonies after 2 days at 30 °C.

The acidity of tarhana dough was determined according to standard TS2282 [18] during fermentation with 0.1 N NaOH to neutralize free acids in 10 g of tarhana expressed as “acidity value”. The pH of dough samples was carried out using a pH meter (Eutech Instruments, Singapore) after 25 mL of distilled water was added to a 5 g sample and homogenized using a WiseStir mixer (HS-50A, DAIHAN, Daegu, Korea).

### 2.8. Monitoring the LAB Diversity in Tarhana Dough Fermented with Bacteriocin Producers

Polymerase chain reaction-denaturated gradient gel electrophoresis (PCR-DGGE) was used to monitor the LAB diversity in the fermentation tarhana dough samples inoculated with *P. acidilactici* PFC69 and *L. lactis* PFC77 as well as the control. Samples were collected during the fermentation (0, 1, 5, 10, 15, and 21 days) and bacterial genomic DNA was isolated from the dough according to the method of Picther et al. [19]. The partial V3 region of 16S rDNA of LAB was amplified from the relevant genomic DNA using the P338 forward primer (5-ACTCCTACGGGAGGCAGCAG-3′) having the GC clamp at the 5′ terminal and P518 reverse primer (5′-ATTACCGCGGCTGCTGG-3′). The PCR conditions were 95 °C for 5 min initial denaturation, 30 cycles of 95 °C for 30 s, 55 °C for 45 s, 72 °C for 1 min, and 72 °C for 10 min final extension. The amplified PCR fragments were run on a polyacrylamide gel with a 25–50% gradient formed with urea and formamide in gradient formed (Bio-Rad, Hercules, CA, USA) in TAE at a constant temperature 60 °C with 50 V for 15 min initially and continued at 150 V for 4 h. Gels were stained with ethidium bromide (Sigma-Aldrich, St. Louis, MO, USA) and visualized on UV 260 nm [20,21].

### 2.9. Statistical Analysis

All the analysis was carried out with at least three replications. The microbiological differences between the tarhana dough samples fermented with/without bacteriocin producers and inoculated with/without indicator strains as well as the control that was fermented spontaneously were determined at each fermentation time-point with a one-way ANOVA analysis of variance using the MINITAB 17.1.0 program (State College, PA, USA).

## 3. Results

### 3.1. Antimicrobial Activity Spectrum of P. acidilactici PFC69 and L. lactis PFC77

*P. acidilactici* PFC69 showed antimicrobial activity against all tested indicators on agar spot as well as inhibited all vancomycin-resistant *E. faecium* (VRE) and five methicillin-resistant *S. aureus* (MRSA). The antimicrobial activity was also detected in supernatants of *P. acidilactici* PFC69 in the well diffusion assay except against Gram-negative rods and MRSA (Table 1). *L. lactis* PFC77 could also inhibit *Micrococcus luteus* DSM1790 and Listeria monocytogenes ATCC 7644 at high level and *S. aureus* ATCC 29213, *S. aureus* ATCC 25923, *B. cereus* ATCC 11778, and *Enterococcus faecalis* ATCC 19443 at moderate levels in both agar spot and well diffusion assays. This strain showed an inhibitory effect against eight of the VRE and two of the MRSA on agar spot, whereas its supernatant was not able to inhibit any VRE and MRSA in the well diffusion assay (Table 1). Accordingly, *P. acidilactici* PFC69 showed wider inhibitory spectrum than *L. lactis* PFC77 in both the agar spot and well diffusion assays.

### 3.2. The pH, Heat, and Enzyme Sensitivity Characteristics of Bacteriocin-Like Substances (BLS) Produced by P. acidilactici PFC69 and L. lactis PFC77

*P. acidilactici* PFC69 and *L. lactis* PFC77 showed 12,800 AU/mL antimicrobial activity in the culture supernatants. Table 2 shows that the inhibitory activity of supernatants harvested from both strains increased at acidic conditions and conversely decreased in alkaline conditions. Both the supernatants maintained antimicrobial activity even at 121 °C without any activity loss at PFC69, but 50% loss at PFC77. However, the proteolytic enzymes degraded the BLS of *P. acidilactici* PFC69 and *L. lactis* PFC77. Only the pepsin did not abolish the activity of BLS of PFC77. All the other enzymes did not have any effect on the antimicrobial activity of BLS of *P. acidilactici* PFC69 and *L. lactis* PFC77. These results show that *P. acidilactici* PFC69 and *L. lactis* PFC77 produced bacteriocins with characteristics such as heat stability, proteolytic sensitivity, and solubility in acidic conditions.

### 3.3. Purification and Molecular Sizes of PFC69 and PFC77 Bacteriocins

Bacteriocins produced by *P. acidilactici* PFC69 and *L. lactis* PFC77 were purified in three steps: precipitation with ammonium sulfate, solid phase extraction, and separated by High Pressure Liquid Chromatography (HPLC) purification. A summary of the activities of bacteriocins in each purification step is shown in Table 3. The antibacterial activity for PFC69 bacteriocin was detected within the peak at 41 min and for PFC77 bacteriocin at 43 min in the HPLC system. The molecular sizes of bacteriocins were determined as approximately 4.5 and 3.5 kDa, respectively, with the tricine-SDS-PAGE system (Figure 1).

### 3.4. Partial DNA Sequences of Bacteriocin Production Encoding Genes in the Genome of P. acidilactici PFC69 and L. lactis PFC77

Genes responsible for the production of bacteriocins produced by *P. acidilactici* PFC69 and *L. lactis* PFC77 strains were identified by extensive PCR screening using primers designed on 22 separate bacteriocin essential genes frequently reported in LAB [17]. Accordingly, DNA fragments corresponding to the *pap*A and *ped*A genes in the genome of the PFC69 strain and nisZ and lcnB genes in the genome of the PFC77 strain were detected. These results showed that *P. acidilactici* PFC69 was a potentially pediocin producer, while *L. lactis* PFC77 could be a nisin and lactococcin producer. It was found that the amplified fragments were 100% homologous with the *nis*Z, *pap*A, and *ped*A genes (Figure 2). However, purine–purine (A–G) transition and two thymine insertions were detected at two points of the *lcn*B fragment. These results indicate that *L. lactis* PFC77 contains two distinct bacteriocin production genes where *lcn*B might be inactive due to having mutations aside from producing nisin.

### 3.5. Microbiological and Chemical Properties of Tarhana Dough Produced with Bacteriocin Producers

The LAB and TAMB amounts increased in all tarhana dough until the fifth day of fermentation and remained stable (Figure 3a,b). In order to determine the effect of bacteriocin producer *P. acidilactici* PFC69 and *L. lactis* PFC77 on *B. cereus* ATCC 11778 and *S. aureus* ATCC 29213, these strains were inoculated to tarhana dough and their growth were observed during fermentation. The amount of *B. cereus* in tarhana dough produced without the bacteriocin producer was higher (*p* < 0.05) at the end of fermentation than when the bacteriocin producer was inoculated, where the *B. cereus* was below detectable level (Figure 3c). Similarly, *S. aureus* growth was inhibited in bacteriocin including the tarhana dough samples. However, the amount of *S. aureus* was high (*p* < 0.05) in the samples where bacteriocin producers were not inoculated and in the control samples fermented with their own microbiota (Figure 3d). These results clearly show that the bacteriocin producers were able to cease the growth of *B. cereus* and *S. aureus* in tarhana fermentation.

The pH variation of all tarhana samples produced with or without bacteriocin producers was similar during fermentation. A rapid pH decrease was observed in the tarhana dough samples in the first few fermentation days, then the pH drop slowed down until the tenth day. After pH 4, there was no longer any considerable change. The acidity of tarhana samples increased during day 15 of fermentation in contrast to a pH decrease and remained stable until the end of fermentation. There was no significant difference among the tarhana dough for acidity level.

### 3.6. The LAB Diversity of Tarhana Fermentation

A wide range of LAB diversity was observed at the beginning of fermentation of the control tarhana dough fermented with spontaneous microbiota. Some new bands were appeared after the fifth day of fermentation, in addition to the existing bands, showing that tarhana was fermented with complex LAB diversity (Figure 4a). When the bacteriocin producers *P. acidilactici* PFC69 and *L. lactis* PFC77 was inoculated, the majority of diversity was maintained during the fermentation with slight loss. Additionally, the both bacteriocin producers were able to be observed during the fermentation. In particular, the corresponding bands of bacteriocin producers became intensified after the fifth day of fermentation (Figure 4b–e). These results clearly showed that the relevant bacteriocin producers did not show any adverse effect on the LAB diversity and sustained their viability during the fermentation of tarhana.

## 4. Discussion

In this study, two bacteriocin producers originating from tarhana fermentation [5] were characterized for activity against the pathogens*: S. aureus* ATCC 29213 and *B. cereus* ATCC 11778. Results showed that *P. acidilactici* PFC69 and *L. lactis* PFC77 were the two bacteriocin producers first characterized from tarhana fermentation. The antimicrobial activity of these two strains was approximately similar to the level of antimicrobial activity detected in the same species [22,23,24,25]. Due to methodological differences in terms of antimicrobial activity, it is not possible to compare the antimicrobial activity of strains exactly [9]. However, it can be asserted that both strains reported in this study had strong antimicrobial activity. In particular, *P. acidilactici* PFC69 inhibited an effect on 48 out of 50 VREs. This result is remarkable in the struggle against the accelerating emergence of bacterial antibiotic resistance [26]. Analysis of the characteristics of antimicrobial activity indicated that the antimicrobial metabolite was bacteriocin due to its sensitivity to proteases. In addition, the molecular size of the purified bacteriocins as well as the gene sequences proved that the *P. acidilactici* PFC69 and *L. lactis* PFC77 strains produced pediocin and nisin-like bacteriocins, respectively. Although these bacteriocins and producers have been characterized by isolating them from different food matrices previously [27,28], these strains are important for belonging to a traditional fermented food like tarhana. Tarhana is a fermented food product containing a complex microbiota including LAB and yeast [3,4,5], and our results show that capable bacteriocin producers might co-exist among members of this microbiota.

Recent advances on whole genome analysis have shown that some *Lactococcus* sp. strains may include multiple bacteriocin production genetic elements [29,30]. In our study, the structural gene screening also showed that *L. lactis* PFC77 contains the *lcn*B gene aside from *nis*A. However, further DNA sequence analysis showed that lactococcin production was inactive in this strain. However, this strain has significance for protective culture development if lactococcin production can be recovered. This is the first report that *L. lactis* contains genetic materials of both nisin and lactococcin production.

Tarhana, a traditional fermented food, is important in nutrition since it contains many nutrients from cereal, milk, and vegetable sources [1]. Although it is an acidic food product, it may have microbiological risks for various pathogens [30]. There was no significant difference for LAB and TAMB counts during fermentation when the *P. acidilactici* PFC69 and *L. lactis* PFC77 strains were inoculated into tarhana dough, however, these strains were able to significantly (*p* < 0.05) inhibit both *B. cereus* ATCC 11778 and *S. aureus* ATCC 29213 growth after the tenth day of fermentation. The significant lower amounts of pathogens detected in dough samples fermented with bacteriocin-producing strains compared to dough samples with only the pathogen inoculated demonstrate that this outcome is related with the bacteriocin production. Since these bacteriocin producers are effective in tarhana dough, these strains show a protective culture feature. In this respect, these strains could be a solution for preventing the microbiological risk for *B. cereus* due to the ingredients used such as flour and raw materials [12].

Some LAB is beneficial in ensuring food safety with bacteriocin production either by the addition of producers or bacteriocins [31,32,33]. It has been shown many times that synthesized bacteriocins are effective in vitro [34]. However, more evidence is required for to understand the effect of bacteriocins in food systems with challenging stress conditions such as competitive microbiota, insufficient nutrients, and adverse environmental conditions [6,35]. In this study, pediocin and nisin like bacteriocin producers, *P. acidilactici* PFC69 and *L. lactis* PFC77, successfully showed protective effects in tarhana with drastic competitive microbiota and stress conditions, and therefore pointed out that these bacteria could be successfully used in cereal-like or complex structural food systems. Thus, these results provide evidence that bacteriocins are one of the important tools in food preservation.

One of the major concerns on bacteriocin usage in the fermentation of tarhana was the possibility of their antimicrobial activity against closely related strains, which may result in inhibition in non-starter LAB or removing the existing microbiota. Additionally, the other question was whether these bacteriocin producers could survive or may compete with the existing microbiota. Our culture independent results indicate that the application of both bacteriocin producers *P. acidilactici* PFC69 and *L. lactis* PFC77 did not result in any significant change in the fermentation microbiota, which is important for maintaining the natural microbiota in the fermentation of tarhana. This might be an indication of interspecies adaptation of microbiota in tarhana fermentation.

## 5. Conclusions

In this study, the *P. acidilactici* PFC69 and *L. lactis* PFC77 strains showed a bioprotective culture feature with substantial pediocin and nisin like bacteriocin production. In particular, the strong antimicrobial activity of *P. acidilactici* PFC69 against VREs highlights this strain in the struggle against the increasing antibiotic resistance of bacteria. In addition, *L. lactis* PFC77 has the potential for strain development studies since it includes both a nisin and lactoccocin production genetic basis. The ability of *P. acidilactici* PFC69 and *L. lactis* PFC77 to inhibit the growth of *B. cereus* ATCC 11778 and *S. aureus* ATCC 29213 in the fermentation of tarhana promises that bacteriocin producers are a significant tool for food preservation.

## Figures and Tables

**Figure 1 microorganisms-08-01083-f001:**
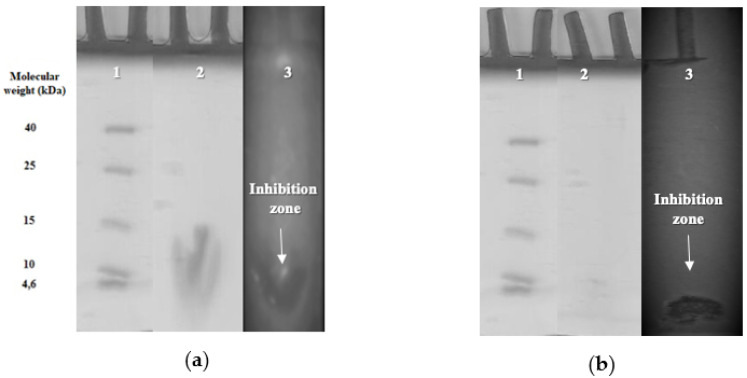
Tricine-SDS-PAGE profile of bacteriocins; (**a**) PFC69 and (**b**) PFC77. Lines: (1) marker, (2) bacteriocin, (3) inhibition zone.

**Figure 2 microorganisms-08-01083-f002:**
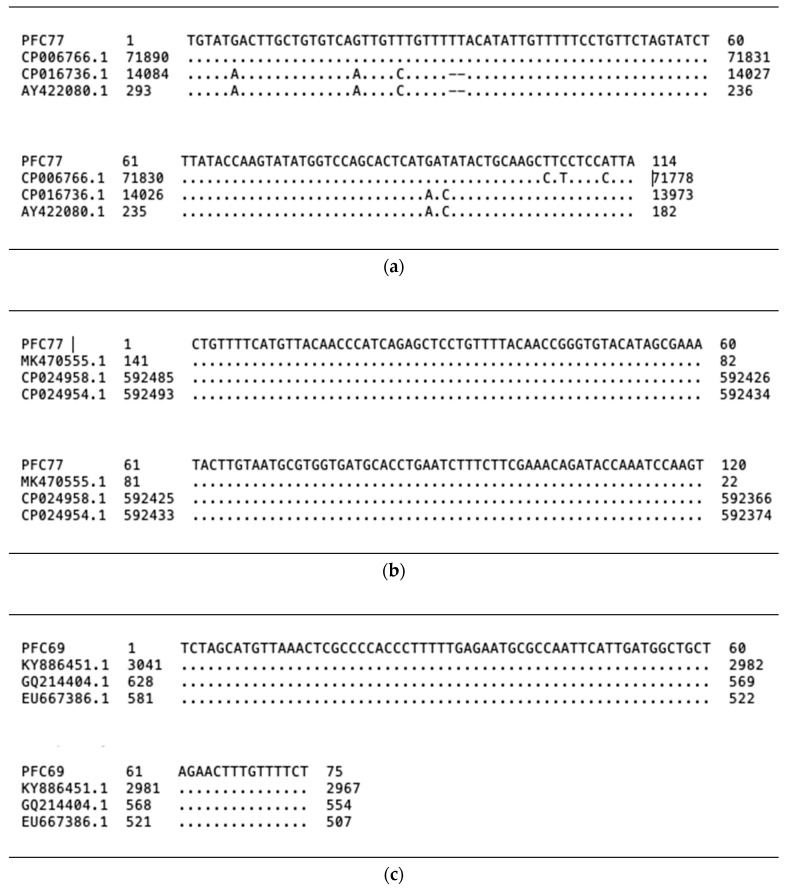
Sequences and homology of bacteriocin encoding genes (**a**) *lcn*B, (**b**) *nis*Z, and (**c**) *ped*A, amplified from the genomes of *P. acidilactici* PFC69 and *L. lactis* PFC69.

**Figure 3 microorganisms-08-01083-f003:**
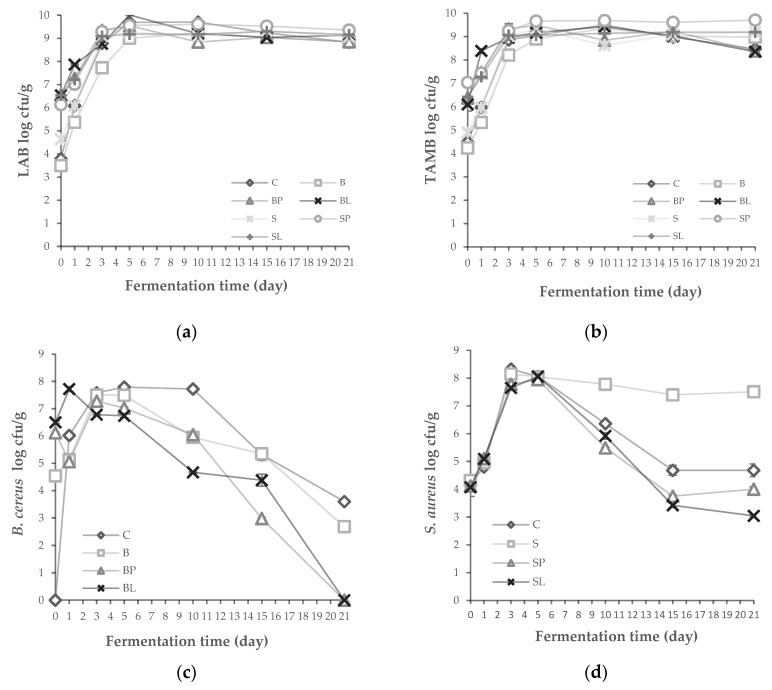
The microbial changes in (**a**) LAB, (**b**) TAMB, (**c**) *B. cereus*, and (**d**) *S. aureus* in the tarhana samples during fermentation produced by inoculating bacteriocin producers and indicator strains. Tarhana dough samples: (C) Control; (B) Inoculated *B. cereus* ATCC 11778 10^4^ cfu/g, (BP) Inoculated *B. cereus* ATCC 11778 10^4^ cfu/g, *P. acidilactici* PFC69 10^6^ cfu/g; (BL) *B. cereus* ATCC 11778 10^4^ cfu/g, *L. lactis* PFC77 10^6^ cfu/g; (S) Inoculated *S. aureus* ATCC 29213 10^4^ cfu/g; (SL) Inoculated *S. aureus* ATCC 29213 10^4^ cfu/g, *L. lactis* PFC77 10^6^ cfu/g; (SP) *S. aureus* ATCC 29213 10^4^ cfu/g, *P. acidilactici* PFC69 10^6^ cfu/g.

**Figure 4 microorganisms-08-01083-f004:**
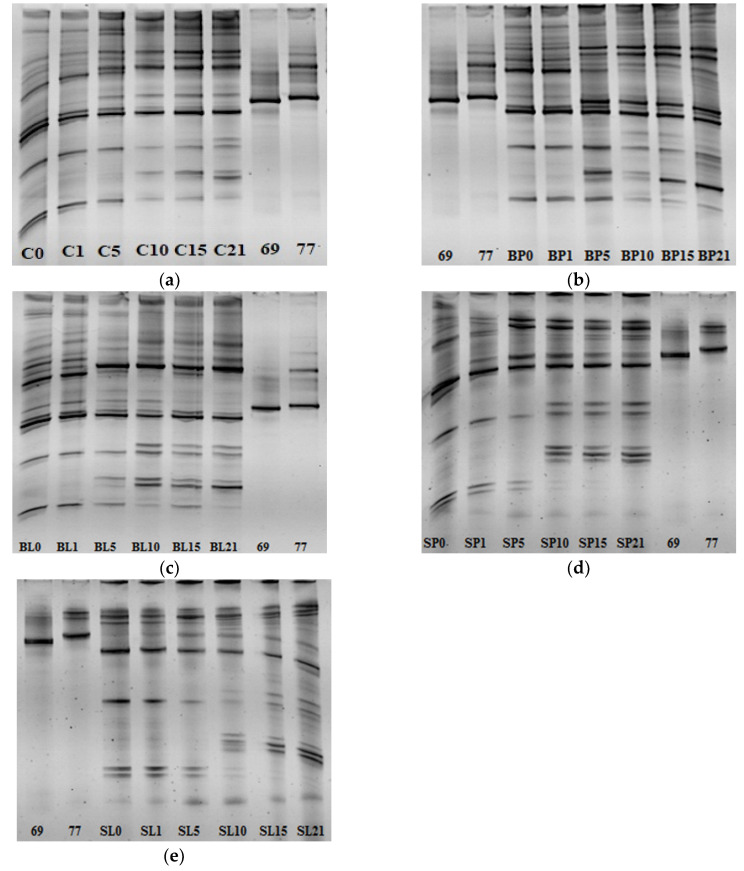
The LAB diversity during the tarhana dough fermentation. (**a**) Control dough fermented spontaneously; (**b**) BP dough fermented with *B. cereus* ATCC 11778 and *P. acidilactici* PFC69; (**c**) BL dough fermented with *B. cereus* ATCC 11778 and *L. lactis* PFC77; (**d**) SP dough fermented with *S. aureus* ATCC 29213 and *P. acidilactici* PFC69; (**e**) SL dough fermented with *S. aureus* ATCC 29213 and *L. lactis* PFC77. The numbers with code correspond to the fermentation day.

**Table 1 microorganisms-08-01083-t001:** The antimicrobial spectrum of *P. acidilactici* PFC69 and *L. lactis* PFC77.

Indicator Strains	*P. acidilactici* PFC69		*L. lactis* PFC77	
	Agar Spot *	Well Diffusion	Agar Spot	Well Diffusion
*M. luteus* DSM1790	+++	+++	+++	+++
*L. monocytogenes* ATCC 7644	+++	+++	+++	+++
*S. aureus* ATCC 29213	++	+	++	+
*S. aureus* ATCC 25923	+++	+	++	+
*E. coli* ATCC 25922	++	-	-	-
*S. typhimurium* ATCC 14028	+++	-	-	-
*B. cereus* ATCC 11778	+	++	++	+
*E. faecalis* ATCC 19433	+++	+++	++	++
VRE *	50/50	48/50	8/50	0/50
MRSA *	5/50	0/50	2/50	0/50

* Inhibition zone diameter: - = <1 mm (no effect), + = 1–5 mm (low effect), ++ = 5–10 mm (medium effect), +++ = > 15 mm (high effect), *: 50 different vancomycin-resistant *E. faecium* (VRE) and methicillin-resistant *S. aureus* (MRSA) isolates. Inhibited isolates/Total isolates.

**Table 2 microorganisms-08-01083-t002:** The pH, heat, and enzyme sensitivity characteristics of bacteriocin-like substances (BLS) produced by *P. acidilactici* PFC69 and *L. lactis* PFC77.

	Isolates	PFC69	PFC77
	Treatments	Activity (AU)	
	Control	12,800	12,800
pH treatments	2	25,600	25,600
	3	25,600	25,600
	4	25,600	25,600
	5	12,800	25,600
	6	12,800	12,800
	7	12,800	12,800
	8	6400	6400
	9	3200	3200
	10	1600	1600
	11	0	800
Heat treatments	80 °C, 5 min	12,800	12,800
	80 °C, 10 min	12,800	12,800
	80 °C, 15 min	12,800	12,800
	90 °C, 5 min	12,800	12,800
	90 °C, 10 min	12,800	12,800
	90 °C, 15 min	12,800	12,800
	100 °C, 5 min	12,800	6400
	100 °C, 10 min	12,800	6400
	100 °C, 15 min	12,800	6400
	121 °C, 15 min	12,800	6400
Enzyme treatments	Catalase	12,800	12,800
	Lysozyme	12,800	12,800
	α Amylase	12,800	12,800
	Lipase	12,800	12,800
	Proteinase K	0	0
	Trypsin	0	0
	Pepsin	0	12,800
	α Chymotrypsin	0	0

**Table 3 microorganisms-08-01083-t003:** Bacteriocin activities produced by *P. acidilactici* PFC69 and *L. lactis* PFC77 in the purification steps.

Isolates	Purification Steps	Volume (mL)	Antibacterial Activity (AU/mL)	Antibacterial Activity (Total AU)
PFC69	Cell free-culture supernatant	500	12,800	6.4 × 10^6^
	Ammonium sulfate precipitation	50	25,600	1.3 × 10^6^
	Solid phase extraction	5	12,800	6.4 × 10^4^
	High-pressure liquid chromatography	1	800	8.0 × 10^2^
PFC77	Cell free-culture supernatant	500	12,800	6.4 × 10^6^
	Ammonium sulfate precipitation	50	25,600	1.3 × 10^6^
	Solid phase extraction	5	800	4.0 × 10^3^
	High-pressure liquid chromatography	1	400	4.0 × 10^2^

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
