# Peer review of "Characterization of Pediococcus acidilactici PFC69 and Lactococcus lactis PFC77 Bacteriocins and Their Antimicrobial Activities in Tarhana Fermentation"

_microorganisms, 2020, doi:10.3390/microorganisms8071083_

Round 1
Reviewer 1 Report
In this manuscript, the authors characterized the metabolites responsible for antibacterial activity of Pediococcus acidilactici PFC69 and Lactococcus lactis PFC77 strains obtained from tarhana. Thus, their antibacterial effect against B. cereus ATCC11778 and S. aureus ATCC29213 was identified during fermentation process. In addition, the stability of PFC69 and PFC77 strains supernatants was tested at different conditions (temperature, pH, etc.). Results, suggested that the identified antimicrobial metabolites are namely bacteriocins. The produced bacteriocins were further purified and their molecular sizes was determined.
This manuscript is well structured and well written. The results are of particular importance to researchers in food chemistry and microbiology fields. Nevertheless, some questions were raised in order to improve the quality of this manuscript. A serious lack of any statistical analysis in this manuscript was observed. Discussion and conclusion section need to be highly improved after conducting a clear statistical analysis.
- Line 10 : Where this product is produced and sold (please specify the location)
- Line 36: please detail why it is critical for individual with weak immunity and add you references
- Line 39: LAB increases ….yeasts are mainly responsible
- Line 8-11 in page 6: These are considered as discussion and not results
- Line 11-12 and line 14-17 in page 7: These are considered as discussion. Please do not discuss any of your results in the results section
- Line 13-14 in page 8: These are considered as discussion. Please do not discuss any of your results in the results section
- Same for section 3.6: Any discussion of the results should be kept for the discussion section. Please make the appropriate modification for all the results section
- Results: Statistical tests should be performed in order to compare your results. Error bars should be present on your figure (Figure 3). In addition, the number of replicates should be well stated.
- Figure 2: is not readable. Higher image resolution is needed
- Figure 4: Figures are extremely small and not well taken
- Discussion: Results not well discussed and compared. This section should be extended and results should be more compared. Comparison of results should be based on statistical differences.
- Conclusion: very short and should be extended further
Author Response
We thank to the reviewer for fruitful comments which improved our manuscript. We tried to do our best to obey all the comments raised.
We thank to the reviewer for fruitful comments which improved our manuscript. We tried to do our best to obey all the comments raised.
- Line 10 : Where this product is produced and sold (please specify the location)
Response 1. Thank you. This is a traditional product produced and consumed in Anatolia but different versions of tarhana have been also produced in Greece, Hungary, Finland. Now we added the necessary information to the “Introduction” part instead of abstract which the reviewer pointed out due to the word limitation of abstract section. Please see the line 32.
- Line 36: please detail why it is critical for individual with weak immunity and add you references
Response 2. Thank you for emphasizing the importance of tarhana for especially infants. This traditional food is prepared with using wheat flour, some vegetables and yogurt which means that it includes both plant and animal nutritional resources. Accordingly, we improved the sentence and added the reference. Please see the line 35.
- Line 39: LAB increases ….yeasts are mainly responsible
Response 3. Revised accordingly.
- Line 8-11 in page 6: These are considered as discussion and not results
Response 4. Yes, you are right. But sometimes we think that short discussion would be useful for the followers. According to your comment, we transformed these discussion sentences to the discussion part. Please see the discussion part.
- Line 11-12 and line 14-17 in page 7: These are considered as discussion. Please do not discuss any of your results in the results section
Response 5. Revised and transformed to the discussion part.
- Line 13-14 in page 8: These are considered as discussion. Please do not discuss any of your results in the results section
Response 6. Revised and transformed to the discussion part.
- Same for section 3.6: Any discussion of the results should be kept for the discussion section. Please make the appropriate modification for all the results section
Response 7. We kindly ask to the reviewer to remain these sentences in the results part. Because these sentences are the justifications to explain why those experiments carried out.
- Results: Statistical tests should be performed in order to compare your results. Error bars should be present on your figure (Figure 3). In addition, the number of replicates should be well stated.
Response 8. The error bars had been included previously but due to figure size and the low range of standard deviation, it is hard to visualize. All the experiments were done with three replications. Now we did statistical comparison at each time point between samples. Please see the line 233. Also see the lines 309, 312, 385.
- Figure 2: is not readable. Higher image resolution is needed
Response 9. Thank you for this comment. Now we re-sized the figure 2 and hopefully it can be readable. But I also would remind that we tried to fit the journal size as well. Please see the line 301.
- Figure 4: Figures are extremely small and not well taken
Response 10. Thank you for this comment. We increase the size of the figures which were appear more transparent. Again, the previous size is the journal requirement. Hopefully the editorial will accept higher sizes. Please see the line 343.
- Discussion: Results not well discussed and compared. This section should be extended and results should be more compared. Comparison of results should be based on statistical differences.
Response 11. Thank you for the critical warnings that gave us chance to think on the results. Thereby we improved the discussion section. Also, we discussed with statistical considerations. Please see the discussion.
- Conclusion: very short and should be extended further
Response 12. The conclusion part was re-written and extended. We added new conclusions relative with the results. Please see the conclusion.
Reviewer 2 Report
The authors adressed a very interesting topic. The methodology is generally well designed, but there are some issues that have to be adressed:
Abstract: what kind of risk for B. cereus/S. aureus? Contaminatio, favorig conditions for bacterial growth or for toxin production?
p1 36-37. A risk not only for the weak immunity - a Staphylococcus enterotoxin or B. cereus toxins can harm also immune competent individuals.
p2 8-13 - please detail in a phrase the pathogenicity mechanisms of S.aureus and B. cereus regarding the food poisoning.
p2 18 - even IF the vast majority.... Also, please state a phrase about the persistence of toxins in food products, even if bacteria are not viable anymore
p2 27 - describe the indicator strains
p2 section 2.2 - you talk about M17G agar plates, but in 2.1 you mention liquid media.
p2 - agar spot method - it is not clear how inhibition zones were formed once the 2nd layer was poured.
p2 43 - which isolates?
p3 2.6 - Describe the extraction method and the primers list
p4 2.4 - The first paragraph need to be reorganized. Punctuation is confusing, looks like ; and : are not used properly.
Table 1 - nothing is mentioned in methods about the 50 clinical isolates.
In figure 1 it is not clear what the inhibition zone represents in SDS-PAGE. No reference to this is mentioned in mathods (2.5). In fig. 1a there is a smear at 4-15 kda, so not sure if there are protein bands.
Bacterial names should be written with italics.
English needs revision (example - see 3.5 first paragraph: S.aureus was high amount...; able to ceased...)
Author Response
We thank to the reviewer for fruitful comments which improved our manuscript. We tried to do our best to obey all the comments raised.

Round 2
Reviewer 1 Report
The author well revised the manuscript and no further modifications are requested from my side. This manuscript may be accepted for publication.
Author Response
thank you so much for your contribution